# Supporting the Frontlines: A Scoping Review Addressing the Health Challenges of Military Personnel and Veterans

**DOI:** 10.3390/healthcare11212870

**Published:** 2023-10-31

**Authors:** Abdullah Alruwaili, Amir Khorram-Manesh, Amila Ratnayake, Yohan Robinson, Krzysztof Goniewicz

**Affiliations:** 1Department of Emergency Medical Services, College of Applied Medical Sciences, King Saud bin Abdulaziz University for Health Sciences, Al Ahsa 36428, Saudi Arabia; 2King Abdullah International Medical Research Center, Al Ahsa 36428, Saudi Arabia; 3Ministry of National Guard—Health Affairs, Al Ahsa 36428, Saudi Arabia; 4School of Health, University of New England, Armidale, NSW 2350, Australia; 5Department of Surgery, Institute of Clinical Sciences, Sahlgrenska Academy, University of Gothenburg, 413 45 Goteborg, Sweden; amir.khorram-manesh@surgery.gu.se; 6Centre for Disaster Medicine, University of Gothenburg, 405 30 Gothenburg, Sweden; yohan.robinson@gu.se; 7Gothenburg Emergency Medicine Research Group (GEMREG), Sahlgrenska University Hospital, 413 05 Goteborg, Sweden; 8Department of Surgery, Army Hospital Colombo, Colombo 00800, Sri Lanka; amila.rat@gmail.com; 9Swedish Armed Forces Centre for Defence Medicine, 426 05 Västra Frölunda, Sweden; 10Department of Security, Polish Air Force University, 08-521 Deblin, Poland; k.goniewicz@law.mil.pl

**Keywords:** combat injury, environmental exposure, healthcare accessibility, military personnel, post-traumatic stress disorder, psychological stressors, traumatic brain injury, veterans

## Abstract

(1) Background: Military personnel and veterans meet unique health challenges that stem from the complex interplay of their service experiences, the nature of warfare, and their interactions with both military and civilian healthcare systems. This study aims to examine the myriad of injuries and medical conditions specific to this population, encompassing physical and psychological traumas. (2) Methods: A scoping review (systematic search and non-systematic review) was performed to evaluate the current landscape of military healthcare. (3) Results: A significant change in the injury profile over time is identified, linked to shifts in combat strategies and the integration of advanced technologies in warfare. Environmental exposures to diverse chemical or natural agents further complicate the health of service members. Additionally, the stressors they face, ranging from routine stress to traumatic experiences, lead to various mental health challenges. A major concern is the gap in healthcare accessibility and quality, worsened by challenges in the civilian healthcare system’s capacity to address these unique needs and the military healthcare system’s limitations. (4) Conclusions: This review underscores the need for holistic, integrated approaches to care, rigorous research, and targeted interventions to better serve the health needs of military personnel and veterans.

## 1. Introduction

Military staff face unique challenges, both in terms of occupational injuries during combat and training as well as exposure to environmental hazards when stationed in high-risk areas. These challenges can lead to a myriad of consequences, ranging from physical and mental stress to life-threatening conditions and enduring disabilities [1,2].

While the term “disability” generally refers to impairments that hinder one’s ability to perform everyday activities and interact with the world, it is worth noting that this can encompass challenges in areas such as communication, hearing, learning, mental health, and social engagement. Though it might be tempting to view all individuals with disabilities under a single umbrella, it is crucial to understand the diverse needs and origins of these impairments. For example, the challenges and experiences faced by military service members may differ significantly from those of civilians, emphasizing the importance of specialized care and understanding for this subset of the population. Further research could shed light on these nuanced differences and the unique needs of our military community [3,4].

Evidence of these distinct challenges comes from comparisons between civilian and military patients [5]. For example, military burn patients from Iraq and Afghanistan displayed more severe injuries in terms of total body surface area but had lower mortality rates. Notably, they had more blast trauma-related injuries, pointing to both a different injury pattern and the benefits of better pre-injury physical conditioning and efficient logistics in the military healthcare system [6,7]. The nature of injuries, like traumatic brain injury (TBI), also varies between military veterans and civilians, underscoring the importance of tailored care [8,9].

Regardless of origin, the management of disabilities—whether from birth or occupational exposure, like in military service members—is multifaceted. It spans activity limitation, impairment, and participation restrictions [3,4]. The World Health Organization’s International Classification of Functioning, Disability, and Health (ICF) offers a framework to understand and classify these dimensions, providing tools for assessing overall health and function [4].

From the above discussion, it is evident that military personnel encounter distinct medical challenges necessitating specialized care immediately post-injury and for long-term management. This is further complicated by the integration of military personnel and veterans into civilian healthcare systems, especially in countries with dual healthcare structures. In contrast, Sweden provides a unique example where a single healthcare system serves both civilians and military personnel [10,11,12].

Due to the specialized medical requirements of military personnel, it is essential to differentiate between the immediate medical response following an injury and the prolonged healthcare approaches tailored for veterans and retirees. This article aims to explore and delineate the present landscape of healthcare for military members, spanning these two dimensions, with a particular emphasis on veteran care. The primary research question guiding this investigation is: What is known about veterans’ and military staff’s immediate and long-term medical needs, and if noted, what are the prevailing challenges and strategies?

## 2. Materials and Methods

### 2.1. Study Design and Protocol

This study employed a scoping review according to Mak and Thomas’s work [13]. A scoping review utilizes a systematic and iterative approach, resulting in a knowledge synthesis of the existing or emerging sets of literature on a defined topic [14]. This study aimed to evaluate the current landscape of military and veteran healthcare, with a focus on immediate and long-term healthcare needs without any emphasis on a single medical condition.

We sought a comprehensive understanding that encapsulates the diverse health challenges faced by veterans and military personnel. We acknowledge that certain conditions, like traumatic brain injuries, might be prevalent in the literature due to their significance in this population. However, our goal was to draw a more holistic picture, incorporating a range of health concerns.

The main reason for choosing a scoping review was our desire to map the extent, range, and nature of the existing literature in the field of veteran and military staff healthcare and further identify the possible gaps. Being aware that such a review is not limited to peer-reviewed literature, we followed the steps recommended by Mak and Thomas [13] by creating a team of experts from both military and civilian healthcare. A librarian was consulted before the search, and the team identified and approved the research question.

### 2.2. Search Keywords and Literature Search Strategy

The selection of keywords was based on the study question (i.e., What is known about veterans’ and military staff’s immediate and long-term medical needs?), expecting that any prevailing challenges and strategies would also be revealed. Using the National Library of Medicine, the MeSH term search was used to apply the most relevant keywords [15,16].

***Veteran:*** The most relevant combination appeared to be “military”, which was also a keyword in our search. Other alternatives appeared to be diverse subgroups and organizations, e.g., Marine, etc.***Emergency:*** This keyword appeared to be most relevant to result in reliable hits, including emergency diseases. Other options were all related to emergency activity or special subgroups, e.g., emergency medicine, etc.***Chronic:*** This keyword appeared to be the most relevant as it encompassed all related diseases with a chronic progression, such as respiratory, liver, kidney, and cancer diseases.***Healthcare:*** This keyword encompassed delivery, financing, disparities, and other relevant search results.

The combination of these words represents the main question in this study. The keywords were employed, either alone or in combination, to maximize the result and garner a diverse and relevant pool of literature. In all cases, the keyword was placed in the double quotation mark (“keyword”) to direct and receive desired hits since, otherwise, the number of irrelevant studies would be much higher (Appendix A).

It is worth noting that our chosen keywords, while seemingly specific, were strategically selected to ensure a broad capture of the relevant literature. For instance, ‘chronic’ was included to capture the literature addressing long-term health challenges and was not intended to limit the search to specific conditions. Our objective was to highlight the overarching health implications of chronic conditions for veterans and military personnel without overly focusing on individual diseases.

### 2.3. Databases and Information Sources

After the team reached a consensus about the search strategy and keywords, the Google Scholar search engine was used as the test bed to produce an overview of the field and the topic. Although there are pros and cons to this search engine, several studies have shown that it can be used to collect relevant publications [17,18]. Following the collection of relevant articles from Google Scholar, several reliable medical databases were chosen to search for relevant and eligible articles [19]. This multi-platform approach, using the Google Scholar search engine together with PubMed, Scopus, Web of Science, and CINAHL databases, ensured a comprehensive collection of the pertinent literature.

### 2.4. Search Strings

The following search strings were utilized in the supplementary Google Scholar engine and other databases:

**Google Scholar:** “Veteran” AND “Military” AND “Emergency” AND “Chronic” AND “Healthcare” (AND “System”)**PubMed:** “Veteran” AND “Military” AND “Emergency” AND “Chronic” AND “Healthcare”**Scopus:** “Veteran” AND “Military” AND “Emergency” AND “Chronic” AND “Healthcare”**Web of Science:** “Veteran” AND “Military” AND “Emergency” AND “Chronic” AND “Healthcare”**CINAHL:** “Veteran” AND “Military” AND “Emergency”

In the Google Scholar search string, several keywords, such as “system”, were added and tested to narrow the result of the search. However, as shown in Table 1, the reduction in hits did not significantly influence the outcome. In the latter case, a reduction of 80 articles was noted. However, all 1100 hits without the “system” keyword were reviewed primarily (Table 1).

### 2.5. Eligibility Criteria

The screening process encompasses a review of all available abstracts. If an abstract was not available, a full-text review of the study was obtained. The main reason for doing so, especially in Google Scholar, is the fact that the content of the papers is not always reflected in the title of a published paper. All included papers have to present special dimensions of veterans’ and military staff’s healthcare according to the aims of this study, with no time limitation in the search. Papers were included if they discussed the health burden in general terms, presenting the short- and long-term impacts of emergency and chronic medical conditions on veterans and military staff. Papers that solely discussed a single medical condition without broader implications for the general health of veterans and military staff were excluded [15,16]. In addition, papers discussing the same diagnosis were meticulously studied to include the most compatible studies matching our inclusion and exclusion criteria. In this way, the number of papers discussing the same condition was reduced, resulting in a manageable number of studies to review.

#### 2.5.1. Inclusion Criteria

The inclusion criteria include all peer-reviewed studies in English that present the necessary information based on the keywords and eligibility criteria, such as quantitative, qualitative, and mixed-method studies.

#### 2.5.2. Exclusion Criteria

The exclusion criteria include papers whose contents did not match the study aims, discussed specific cases or special diseases, or were in languages other than English.

### 2.6. Selection of Sources of Evidence

All studies were reviewed by two authors to evaluate the title, the contents, and the abstract. If consensus was not achieved, a third reviewer was asked to make the final decision. The final search results were exported into an Excel file, and two reviewers evaluated the collected papers. Duplicates were removed, ensuring a unique collection of papers. Both reviewers meticulously assessed each included paper for its relevance to our study’s aims, sidelining any off-target literature based on the inclusion and exclusion criteria. Any disagreement was resolved by discussing it with a third reviewer [13].

### 2.7. Review Process and Data Charting

**Primary Review**: The result obtained from the Google Scholar search was initially assessed by one of the authors. Abstracts of 1100 papers were reviewed, and 498 studies were selected for the second round of review. This was possible because hundreds of papers could be found discussing only one diagnosis. For instance, 408 papers discussed brain injuries, and 85 discussed amputations of extremities due to vascular or skeletal injuries in the military context. In dubious cases, related articles or other articles citing the article were also reviewed. Once refined, our collection underwent a non-systematic review, allowing us to dive deep into each paper’s core insights because of the heterogeneity found in the selected articles. The main reason for a non-systematic review was the heterogeneity of the results with several types of articles, preventing a correct evidence evaluation.

While many papers focused on specific diagnoses, we prioritized those that provided broader insights into the general health landscape of veterans and military personnel. This approach ensured our review remained aligned with our initial objective of understanding the comprehensive health needs of this population.

**Secondary Exploration**: In the second review, all papers selected from the supplementary Google Scholar engine and medical databases were reviewed by two authors (Point 2.8). A data extraction form was developed collaboratively for all included papers, including the name of the author, geographical location, study methods and population, main results, and theme in each paper, which are presented in Table 2 [15,20]. Even in this phase of the study, special attention was paid to not including several papers on the same diagnosis.

### 2.8. Content Categorization

Upon gathering the dataset, the theme of each study was determined, and the outcomes were categorized into distinct thematic clusters, providing a structured representation of the data and synthesis of results [15,16]. This categorization aimed to synthesize insights from diverse conditions and challenges into overarching themes. By doing so, we maintained a broader perspective on the health needs of veterans and military personnel, ensuring that our findings were not overly skewed towards any single condition. These thematic sections were: (1) Physical Impacts and Injuries; (2) Environmental Exposure and Impact; and (3) Stressors and Psychological Impacts (see Table 2). In the thematic analysis, texts in each paper were examined to find the link between the text and the research question. A code was created and modified in an iterative process during data analysis, which was a reflective discourse among involved experts [15,35,36]. Since no previous studies have looked into this topic in general terms, a deductive content analysis was chosen in this phase of the study.

## 3. Results

The search results for the Google Scholar search engine and diverse databases are shown in Figure 1. The initial evaluation of the results from each search engine was as follows: Google Scholar (n = 498), PubMed (n = 6), Scopus (n = 8), Web of Science (n = 7), and CINAHL (n = 9). Regarding our findings from the Google Scholar search, we had to evaluate every abstract to identify the most reliable and relevant papers since the number of published papers with the same diagnosis was high. Therefore, the removal of studies was conducted in two phases. In the first phase, clear medical conditions such as brain injury were evaluated, and relevant papers were moved to the second phase while the remaining papers were removed. Of a total of 524 papers, duplicates and irrelevant papers were removed to yield the 17 final papers included in the review. These papers represented the sum of papers illustrating the current landscape of medical challenges for military service members. Figure 1 shows the result of the literature selection according to the PRISMA flow chart. Table 2 shows a summary of the selected literature that was included in this study. Most of the studies were narrative reviews (n = 11), a few were systematic reviews, one was a randomized trial, one was registry-based, and another was a comparative study. Consequently, the evidence-based quality of the included 17 studies differed and did not allow a proper assessment (Appendix A).

### 3.1. Physical Impacts and Injuries

Military personnel are susceptible to a wide range of battlefield injuries and medical conditions. These can include shrapnel and gunshot wounds, limb amputations, and traumatic head and brain injuries sustained during combat. Additionally, many suffer from tinnitus and hearing loss due to noise exposure, as well as sprains, strains, and limited range of motion—particularly in the ankles and knees—resulting from strenuous exercises and missions in challenging environments [37,38].

The rate and type of injuries in warfare have evolved, reflecting shifts in combat strategies and technologies. While both urban warfare and armored warfare have been integral components of military strategy since WWII, their prevalence and the contexts in which they are employed have varied. For instance, the Battle of Bachmut in 2014 in Ukraine showcased the intricacies of urban warfare. Concurrently, various campaigns have highlighted the nuances of armored warfare. Recent analyses from the war science literature illustrate these shifts and their implications for military health [2]. This shift has been bolstered by the integration of advanced technologies, such as drones [39]. While not entirely novel, these technologies have revolutionized modern conflict dynamics. The increasing targeting and involvement of civilians in conflicts, as evidenced in wars such as WWII, Vietnam, and more recent conflicts in Sudan, challenges traditional estimation methods. This trend underscores a concerning deviation from or neglect of the principles outlined in international humanitarian law. Numerous reports and studies have highlighted how conflict parties have frequently disregarded these principles, leading to dire consequences for civilian populations [40,41]. Furthermore, the emergence of hybrid warfare—a blend of conventional, irregular, and cyber warfare tactics, often combined with measures to influence public opinion and political policies—has revolutionized modern conflicts. This transformation underscores the need for a comprehensive approach to assessing not only physical casualties but also the mental health repercussions that arise from such multifaceted combat scenarios. This change demands not only rigorous analysis but also continuous monitoring of military healthcare requirements and casualty records [24,42,43]. Most major military powers and alliances are aware of the potential offered by big data for military health analysis. However, they also recognize the risks associated with this type of data, as it may expose vulnerabilities.

#### 3.1.1. Mechanism and Type of Injuries

Existing research, particularly those analyzing the aftermath of WWII, has shown that approximately 13.6% of the total casualty rate is composed of armored units. Monthly casualty reviews across various army branches have revealed some discrepancies, but only about 55% of the battle casualties were indeed irretrievable losses [44,45]. The reported mechanisms of combat injuries in armored divisions break down as follows: 50% ballistic, 5% blunt, 5% blast, 25% thermal, and 15% combined. This distribution suggests that each conflict may show a distinct casualty and injury profile [46]. The available literature provides insight into injury locations: head and neck (4–31%), thorax (4–27%), abdomen (2–22%), pelvis (1.6–3.8%), and limbs (31–87%) [36,41]. Based on these reports, a rough estimation suggests that “should the primary medical treatment and evacuation system function as planned, the military casualty rate in modern warfare would likely hover around 13–14% in rural settings and around 6–7% in urban areas. The mortality rate could potentially range from 5–10%” [24,46]. Furthermore, there has been a rise in limb injuries, while pelvic injuries remain relatively infrequent. Other injury types, such as those to the head and thorax, fluctuate within a specific range [2,24,38].

#### 3.1.2. Main Injuries and Sequelae of Combat Injuries

The physiological and psychological stressors endured in military service can precipitate various health issues. One notable condition is post-traumatic stress disorder (PTSD), a mental health disorder that may arise after experiencing severe trauma, such as combat scenarios [22]. Unlike acute stress reactions, which are immediate and short-lived responses to traumatic events, PTSD is characterized by persistent symptoms that last for at least one month and interfere with daily functioning. These symptoms encompass flashbacks, nightmares, emotional detachment, and others. Neuroscience studies have linked PTSD to changes in specific brain structures—namely the amygdala, hippocampus, and prefrontal cortex—that play pivotal roles in emotions, memory, and executive functions.

Traumatic brain injury, mild or severe, is another significant health issue among military personnel and veterans and can occur due to explosions, vehicular accidents, or other impactful incidents [27]. TBIs may lead to persistent or even permanent cognitive, physical, and emotional symptoms. Recent research has associated TBI with an accumulation of abnormal proteins such as tau in the brain, which is implicated in neurodegenerative diseases like Alzheimer’s. Consequently, neurosurgical rehabilitation of TBI becomes crucial in assisting patients in regaining cognitive and motor functions, thereby enhancing their overall quality of life [25,27].

With the observed increase in limb injuries, orthopedic rehabilitation has become an essential component of care for military personnel and veterans. This is true for those managing injuries that can be treated conservatively, as well as for individuals who undergo surgical interventions, including amputations. The loss of a limb can be a devastating result of service-related injuries. Specialized rehabilitation services are needed to help individuals regain mobility and independence and adapt to prosthetic devices, not underestimating the need for pain management in both the short and long term [26,47].

Urological care for neurogenic bladders resulting from spinal trauma also poses a significant challenge for service members. This specialized care is pivotal in managing complications such as urinary incontinence and recurrent infections, thereby improving patients’ overall quality of life [48].

Oral and maxillofacial (OMF) care for facial disfigurements is another critical healthcare aspect for military personnel and veterans [49]. This specialized care focuses on reconstructing and rehabilitating facial structures, significantly affecting an individual’s appearance, self-esteem, and abilities to communicate and eat. In addition, diverse aspects of plastic reconstructive work—such as treating soft tissue injuries, skin defects, and pressure sores—are integral to the comprehensive care of military personnel and veterans [50]. These interventions can help restore function, appearance, and comfort to those who have sustained severe injuries during their service.

### 3.2. Environmental Exposure and Impacts

Military service members are at risk of exposure to environmental hazards, such as contaminated water, chemicals, infections, and burn pits [1]. A notable historical instance is the exposure to Agent Orange, a chemical herbicide and defoliant, during the Vietnam War. This exposure has been linked to an increased risk of developing specific types of cancer and other health complications [51]. Despite significant research, comprehensive treatments for these conditions are still elusive, primarily due to the diverse range of health issues associated with this exposure. Further complicating matters is the latency period, which can span decades before symptoms appear and prevent early detection and treatment efforts.

Similarly, Gulf War Syndrome, a chronic, multi-symptomatic disorder affecting military veterans from both sides of the 1990–1991 Persian Gulf War, has been characterized by Bjørklund et al. [52]. The multifaceted nature of this syndrome, characterized by a wide spectrum of symptoms that include fatigue, headaches, cognitive dysfunction, and other systemic manifestations, poses challenges in both diagnosis and management. To date, there is no universally effective treatment for Gulf War Syndrome, largely due to its heterogeneity and the complex interplay of genetic, environmental, and psychological factors.

In both of these examples, the overarching challenge lies in the complexity and diversity of the health issues involved, the long latency periods, and the complex interplay of various risk factors. These factors combined highlight the need for continued research to better understand these conditions and develop more effective approaches to their early detection, diagnosis, and management.

### 3.3. Stressors and Psychological Impacts

Military servicemen might be affected by diverse stressors, such as routine stress, stress caused by sudden changes in their lives, and traumatic stress when in danger of or experiencing serious harm or death [31]. The latter may result in mental health problems such as anxiety, post-traumatic stress disorder, depression, substance use, and suicide. Some studies have discussed the process of family detachment [53,54,55], operational stress [56], and the importance of social support and the availability of special psychological care for the military staff, indicating the prevalent occurrence of diverse psychological conditions due to different existing stressors in military life and the need for preventive and responsive measures [34,57].

Addressing multidimensional health concerns and improving the well-being of military personnel and veterans requires adopting a multifaceted approach. This includes integrating mental health screenings and support services into routine medical care for service members and cultivating an open dialogue about mental health to reduce stigma within the military community. Moreover, further investment in research to better understand the unique health challenges faced by military personnel is crucial, as it can inform the development of targeted treatments and interventions [58,59].

### 3.4. The Current Issues for Military Service Members to Receive Adequate Healthcare

While civilian healthcare systems have their own distinct challenges, military personnel must receive proper care for their unique and often specialized needs. There are distinct differences between the medical concerns typically seen in military staff, especially combat-related injuries and certain chronic conditions, compared to those in the civilian population. Addressing these disparities requires both training and awareness on the part of civilian healthcare providers [38,60]. It is also crucial to recognize that a lack of familiarity with the full spectrum of conditions and psychological concerns affecting military personnel might exist among civilian healthcare staff [26]. However, in both wartime and peacetime, the goal should always be to ensure equal access and quality of care for all, regardless of civilian or military status.

On the other hand, even in those countries that have a separate military healthcare system, several issues appear to be a real obstacle for veterans and military service members to receive adequate care. For instance, in the US, only around half of the existing veterans (over 9 million) are registered to receive healthcare through the Veteran Healthcare Administration [61,62]. Another report from the RAND Center for Military Health Policy Research also shows that only 30% of the 50% of registered veterans in need of mental health services receive proper and evidence-based care [63]. The most important reasons for not receiving adequate healthcare were (1) the complicated healthcare system and the labyrinth of diverse specialties; (2) long wait times; and (3) disparities in rural areas [60]. Although these issues may be similar to what civilians experience, they also add to the already existing struggles that military staff and veterans face in integrating into civilian society [32].

## 4. Discussion

This scoping review underscores the distinct health challenges military service members face due to trauma or environmental hazards compared to civilians. These challenges are defined by unique mechanisms, outcomes, and requisite treatments. As military personnel transition to civilian life, they often encounter the healthcare system’s constraints. While grappling with their specific medical conditions, many also face hurdles such as navigating a healthcare system that can be complex, characterized by prolonged wait times and disparities in access, particularly in rural regions. Addressing both their immediate and long-term medical needs becomes paramount. Despite efforts, gaps remain in how the system accommodates the specific needs of veterans and active-duty personnel, which this review seeks to elucidate [32,61,63].

The healthcare issues that military personnel and veterans grapple with span a wide spectrum, from post-traumatic stress disorder (PTSD) and traumatic brain injury (TBI) to extensive rehabilitation requirements for amputees. This delineates the unparalleled challenges they face, mandating expertise across multiple medical disciplines like neuroscience, orthopedics, and oral and maxillofacial care. The necessity for such a broad spectrum of care accentuates the interlinking nature of their health problems and underscores the urgency for a comprehensive, evidence-backed, and integrated medical approach [22,25,27,34,47,48,49,50].

One crucial avenue to bolstering the understanding of military-specific health challenges lies in the realm of education. By embedding military medicine within medical and allied health curricula, the next generation of healthcare providers can be adequately equipped to address the specialized needs of military personnel and veterans. This integration serves a dual purpose. Firstly, it cultivates a more informed understanding among practitioners about the unique health challenges, trauma mechanisms, and environmental hazards that military personnel encounter. Secondly, it fosters a bridge of communication and trust between healthcare providers and their military patients by demonstrating a deepened understanding of the latter’s experiences and challenges.

As this review underscores the multifaceted health challenges of military personnel, it becomes apparent that there is a strong case for the establishment of a dedicated subspecialty for military medicine within the civilian healthcare framework. A subspecialty dedicated to military medicine would ensure that the niche and highly specialized needs of veterans and active-duty personnel are met with precision and expertise, bridging the knowledge gap that often exists in generic medical practices. Furthermore, such a specialty would act as a nexus, intertwining the specialized medical approaches of the military healthcare system with the broader resources and practices available in civilian healthcare. By doing so, a seamless continuum of care can be achieved, allowing military personnel and veterans to access tailored medical interventions irrespective of their healthcare setting [64].

To serve the needs of veterans, a supportive ecosystem tailored for them is essential. Such an ecosystem must be built on collaboration among military establishments, healthcare providers, and policymakers, particularly in contexts like the US, where the military–veteran population is significant [60,65]. Furthermore, families and local communities play an invaluable role in promoting the well-being of military personnel and veterans [21]. The focus should be on devising strategies that channel resources and support to these key stakeholders. Through such dedicated support systems, we not only prioritize veterans’ well-being but also honor their sacrifices, envisioning a post-service future where they lead fulfilling lives.

A pivotal element in this ecosystem is a healthcare system that seamlessly integrates care, whether it is under a military or civilian umbrella. This integrated approach should encompass various phases of medical treatment. While some integrative efforts can be driven by cost-saving measures, it is crucial to ensure that the primary objective remains the provision of holistic and coordinated care. Achieving true integration requires the confluence of various specialties under a multidisciplinary umbrella focused on delivering patient-centered care [29,66].

### 4.1. Prevention and Early Intervention

The foundation for addressing the multifarious health repercussions military personnel endure is rooted in early preventive measures. It is beneficial to enhance training, instill awareness about potential hazards and preventive tactics, and train personnel to recognize early symptoms [30]. An integral component of this strategy is regular mental health check-ups during and after service, which can significantly ameliorate long-term outcomes for conditions like PTSD and depression.

### 4.2. Specialized Care and Long-Term Management

Optimal care for service-related ailments and the management of chronic conditions hinges on civil–military healthcare collaboration, a nexus of military institutions, healthcare providers, and policymakers. Pioneering treatments and rehabilitation methods, informed by rigorous scientific research, can significantly enhance both physical and psychological outcomes for veterans. A focus on research and customized treatments for military-specific health challenges remains indispensable [67].

### 4.3. Family and Community Support

The linchpin of the entire support structure is often the families and communities of service members [68]. Constructing a robust support matrix for veterans entails resources and support services for these key stakeholders. Potential measures encompass family counseling, peer support groups, and educational initiatives that impart insights into the exclusive challenges service members endure [23].

### 4.4. Reintegration into Civilian Life

Community-driven programs are instrumental in facilitating the smooth transition of military personnel into civilian life. By catalyzing social interactions, skill enhancement, and job support, these initiatives assist veterans in forging fresh connections, redefining their objectives, and adapting to civilian life. Collaborative endeavors that engage local businesses, NGOs, and academic institutions can sculpt an inclusive environment for veterans, fostering a sense of community and simplifying their civilian transition [33,59].

### 4.5. Public Awareness and Advocacy

Elevating societal awareness about the health challenges military personnel face is pivotal. A well-informed society can drive empathy, facilitate supportive policies, and be the catalyst for research in this domain [28]. Moreover, as society becomes more attuned to these challenges, there is a growing recognition of the nuanced and specialized needs of military medicine.

To efficiently manage the myriad complexities in military medicine, there is a case to be made for instituting a subspecialty that holistically addresses the diverse needs and health conditions of military personnel. This specialty could potentially oversee the care of military staff at dedicated military medical centers in countries with military healthcare infrastructures and could operate as a subspecialty within the civilian healthcare realm.

### 4.6. Limitations

The design and constraints of our study predominantly spring from the chosen research methodology. Our selection of a scoping review was due to the fact that we aimed to identify knowledge gaps and scope a body of the literature [69]. In contrast to a systematic review, we did not aim to dive into the details of a diagnosis but to collect general information regarding the care of a special group of the population. To do so, we also followed the steps recommended by Mak and Thomas [13]. However, the speed at which our review was conducted can limit the thoroughness of the literature search, and in turn, this may have led to the omission of key studies, introducing potential gaps in our compiled insights.

Another layer of limitations is the potential bias associated with the use of diverse keywords. Although we examined the MeSH words associated with our search, we know other keywords may have resulted in different search results. Therefore, our selection may have resulted in us possibly missing out on crucial publications, and we may have introduced a selection bias. Moreover, there exists an inherent risk that our review could be skewed toward studies that predominantly report positive or notable outcomes, a reflection of the prevalent publication bias in the academic literature. Our emphasis on publications in English also added a constraint, potentially sidelining meaningful research presented in other languages.

The use of the Google Scholar search engine is associated with pros and cons, as discussed in the method section [17,18,19]. However, as reported by Masterangelo and colleagues [70], it can be used as a complementary search to existing databases, such as PubMed. Despite all these gaps, the use of four different databases and Google Scholar should have resulted in major publications and might have reduced the bias in our search results. Nevertheless, given these considerations, we urge our readers to approach the findings with a degree of circumspection. While we believe our results are insightful, they might not capture the full breadth and depth of evidence on the subject.

Additionally, our reliance on major international databases might have unintentionally sidelined research from regional or less globally recognized databases. This could have introduced a geographic bias, potentially missing out on studies from certain regions or countries. While we aimed for comprehensiveness, this limitation might have led to an underrepresentation of certain perspectives or findings.

## 5. Conclusions

Military personnel, due to their distinct service conditions, face heightened risks from multiple physical, psychological, and environmental hazards. These often result in serious injuries, long-term disabilities, or fatalities. Existing healthcare structures, whether military or civilian, show evident gaps, especially in addressing chronic conditions common among veterans and active military members. The specialized nature of these conditions calls for a holistic approach that is comprehensive and attuned to their specific origins, clinical presentations, and the necessary medical and rehabilitative interventions.

Given the relatively infrequent appearance of these conditions in the civilian populace, it becomes imperative for healthcare professionals to provide this specialized care. This can be achieved either within dedicated military healthcare establishments or through distinct subspecialty units in civilian health infrastructures. Adopting this approach ensures the delivery of rigorous, evidence-based care, fostering a comprehensive support ecosystem that duly acknowledges and respects the sacrifices of military personnel and veterans. This vision aims for their continued well-being in the long term.

For improved results, medical and allied health curricula should embed military medicine components. By learning about this specialty, future healthcare professionals are better poised to comprehend and address the unique health challenges that military personnel and veterans encounter. Furthermore, those specializing in military medicine should predominantly serve within the military healthcare realm. Still, there should be provisions for them to occasionally rotate within civilian healthcare environments. Such rotations can spark an exchange of expertise and knowledge, elevating care standards across both sectors.

Finally, there is a strong rationale for carving out a dedicated subspecialty for military medicine within the civilian healthcare milieu. This subspecialty can function as a conduit, marrying the specialized care methodologies of the military healthcare spectrum with the vast resources and methodologies of civilian healthcare. By doing so, we ensure that both veterans and active-duty staff can avail themselves of bespoke care, irrespective of the healthcare setting they approach.

## Figures and Tables

**Figure 1 healthcare-11-02870-f001:**
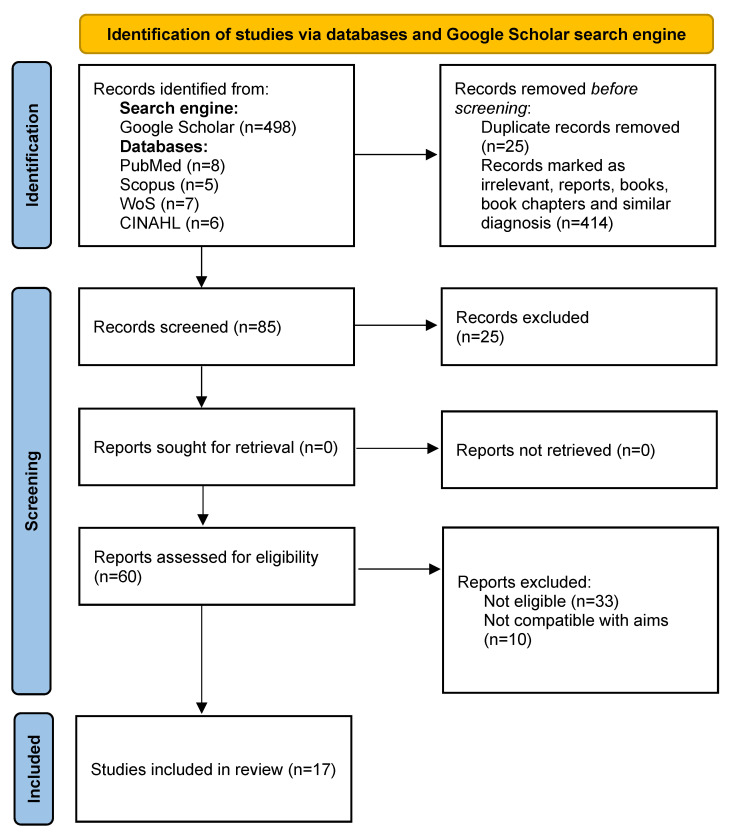
PRISMA flow diagram showing the process of literature selection [14].

**Table 1 healthcare-11-02870-t001:** The outcome of each search engine query.

Search Keywords	Hits
**Google Scholar**	
“Veteran”	52,200
“Veteran” AND “Military”	10,500
“Veteran” AND “Military” AND “Emergency”	2920
“Veteran” AND “Military” AND “Emergency” AND “Chronic”	1640
“Veteran” AND “Military” AND “Emergency” AND “Chronic” AND “Healthcare”	1100
“Veteran” AND “Military” AND “Emergency” AND “Chronic” AND “Healthcare” AND “System”	1020
**PubMed**	
“Veteran”	66,776
“Veteran” AND “Military”	12,495
“Veteran” AND “Military” AND “Emergency”	220
“Veteran” AND “Military” AND “Emergency” AND “Chronic”	31
“Veteran” AND “Military” AND “Emergency” AND “Chronic” AND “Healthcare”	6
**Scopus**	
“Veteran”	18,106
“Veteran” AND “Military”	2770
“Veteran” AND “Military” AND “Emergency”	114
“Veteran” AND “Military” AND “Emergency” AND “Chronic”	21
“Veteran” AND “Military” AND “Emergency” AND “Chronic” AND “Healthcare”	8
**WoS**	
“Veteran”	18,285
“Veteran” AND “Military”	3685
“Veteran” AND “Military” AND “Emergency”	115
“Veteran” AND “Military” AND “Emergency” AND “Chronic”	17
“Veteran” AND “Military” AND “Emergency” AND “Chronic” AND “Healthcare”	7
**CINAHL**	
“Veteran”	5833
“Veteran” AND “Military”	1236
“Veteran” AND “Military” AND “Emergency”	9
“Veteran” AND “Military” AND “Emergency” AND “Chronic”	-
“Veteran” AND “Military” AND “Emergency” AND “Chronic” AND “Healthcare”	-

**Table 2 healthcare-11-02870-t002:** The characteristics of the selected and included studies and the theme for each study. Physical impacts include all injuries in combat, types, and mechanisms. Environmental exposure includes environmental hazards during combat and after retirement. Stressors and psychological impacts include all factors triggering and resulting in subsequent sequelae.

Biography	Method	Main Findings	Conclusions	Theme
Huebner, AJ. (2009). Shadowed by war: Building community capacity to support military families. Family Relations, 58(2), 216–228. [21]	Narrative review	A community capacity model is presented together with programmatic examples to suggest new and viable ways to support military families in the changing context of post-9/11 military service. To support service members, researchers, educators, and policymakers should more effectively continue to explore and refine these concepts.	A community capacity approach has clear merit for strengthening community support systems for military families and service members. This community-oriented approach seeks to connect potentially isolated families to areas of support. Rather than relying strictly on the top-down approach of military-sponsored support, this model provides a framework for helping families to help themselves in the context of their own community.	Family support, as in Stressors and Psychological Impacts
2. Creamer M, (2011). PTSD among military personnel. International Review of Psychiatry; 23(2):160–5. [22]	Narrative review	PTSD is NOT the only mental health condition that may develop because of military experience. PTSD is just one of several possible conditions, with depression and substance use disorders being equally, if not more, common. Nevertheless, the fact that PTSD is etiologically linked to a stressful experience in the diagnostic criteria ensures that the condition will remain of central interest to defense forces around the world.	Despite great progress in preventive medicine, a great deal of learning remains about prevention, early identification, and effective treatment of PTSD among military staff and veterans. The biggest current challenge is ensuring evidence-based interventions are appropriately implemented within the military and veteran service systems.	Psychological stressors, as in Stressors and Psychological Impacts
3. Hazle M (2012). Helping Veterans and Their Families Fight on! Advances in Social Work. 2012;13(1). [23]	Narrative	All military staff come home from war with changes. For some, leaving the combat zone marks the beginning of a new battle—to reconcile the emotional changes that have resulted from their experiences, to smoothly reunite with their families, to put the stresses of combat behind them, and to find employment that fulfills them and uses the skills they have gained.	Many veterans and their family members need expert help rebuilding their lives. To adequately respond to the needs of veterans and their families and strengthen their reintegration into the community, a multidisciplinary operation as a single unit—a collection of capable organizations and individuals, each with specific focus areas and tasks—is needed.	Psychological and family support, as in Stressors and Psychological Impacts
4. Forneris, C. (2013). Interventions to prevent post-traumatic stress disorder: a systematic review. American journal of preventive medicine, 44(6), 635–650. [8]	Systematic review	In three studies for people with acute stress disorder, brief trauma-focused cognitive behavioral therapy was more effective than supportive counseling in reducing the severity of PTSD symptoms (moderate strength); these two interventions had similar results for the incidence of PTSD (low strength), depression severity (low strength), and anxiety severity (moderate strength). PTSD symptom severity after injury decreased more with collaborative care than usual care (single study; low strength). Debriefing did not reduce the incidence or severity of PTSD or psychological symptoms in civilian trauma (low strength).	Evidence is very limited about best practices for treating trauma-exposed individuals. Brief cognitive behavioral therapy may reduce PTSD symptom severity in people with acute stress disorder; collaborative care may help decrease symptom severity post-injury.	Psychological support and treatment, as in Stressors and Psychological Impacts
5. Belmont, PJ. (2013). “The nature and incidence of musculoskeletal combat wounds in Iraq and Afghanistan (2005–2009).” J Orthop Trauma 27(5): e107–113. [24]	Registry	The Joint Theater Trauma Registry (JTTR) contained data on 6092 musculoskeletal casualties with 17,177 wounds. Seventy-seven percent of all casualties sustained a musculoskeletal wound. The incidence of musculoskeletal combat casualties was 3.06 per 1000 deployed personnel per year, with fractures occurring in 3.41 per 1000 and soft-tissue wounds occurring in 4.04 per 1000. Amputations represented 6% of all combat wounds. Most musculoskeletal wounds were caused by explosive blasts (*p* < 0.001), as were nearly all traumatic amputations.	This study represents the most complete description of the scope of orthopedic war trauma. It also presents injury-specific incidences that have not previously been described for musculoskeletal combat casualties. Musculoskeletal casualties may occur in 3 of every 1000 personnel deployed per year.	Physical injury rehabilitation, as in Physical Impacts and Injuries
6. Brasure, M. (2013). Participation after multidisciplinary rehabilitation for moderate to severe traumatic brain injury in adults: a systematic review. Archives of physical medicine and rehabilitation, 94(7), 1398–1420. [25]	Systematic review	This article presents the results of selected key questions from a recent Agency for Healthcare Quality and Research comparative effectiveness review. Twelve studies met our inclusion criteria; of these, 8 were at low or moderate risk of bias (4 randomized controlled trials of 680 patients and 4 cohort studies of 190 patients, sample size 36–366). Heterogeneous populations, interventions, and outcomes precluded pooled analysis. Evidence was insufficient to conclude effectiveness. Evidence on comparative effectiveness often showed that improvements were not different between groups; however, this evidence was of low strength and may have limited generalizability.	This review focused on participation in multidisciplinary rehabilitation programs for impairments ranging from moderate to severe TBI. The available evidence did not show the superiority of one approach over another. This conclusion is consistent with earlier reviews that examined other patient-centered outcomes. While these findings will have little clinical impact, they do point out the limited evidence available to assess effectiveness and comparative effectiveness while highlighting important issues to consider in future comparative effectiveness research on this topic.	Multidisciplinary rehabilitation, as in Physical Impacts and Injuries; Stressors and Psychological Impacts
7. Vallerand, A. (2015). Pain management strategies and lessons from the military: A narrative review. Pain Research and Management, 20, 261–268. [26]	Review	Wounded soldiers often experience substantial pain, which must be addressed before returning to active duty or civilian life. This study provides a narrative review regarding US military pain management guidelines and initiatives that may guide improvements in pain management, particularly chronic pain management and prevention, for the general population.	The application of US military pain management guidelines has been shown to improve pain monitoring, education, and relief. Inadequate pain management, particularly inadequate chronic pain management, is still a major problem for the general population in the US. Application of military strategies for pain management to the general US population may lead to more effective pain management and improved long-term patient outcomes.	Pain management rehabilitation, as in Physical Impacts and Injuries; Stressors and Psychological Impacts
8. Moriarty H. (2016). A randomized controlled trial to evaluate the veterans’ in-home program for military veterans with traumatic brain injury and their families: report on impact for family members. Pm&r. ;8(6):495–509. [27]	Randomized controlled trial	Traumatic brain injury creates many challenges for families as well as for patients. This study used an innovative intervention for veterans with TBI and their families—the Veterans’ In-home Program (VIP)—targeting veterans’ environments, delivered in veterans’ homes, and involving their families. The study determines whether the VIP is more effective than standard outpatient clinic care in improving family members’ well-being in 3 domains (depressive symptoms, burden, and satisfaction) and assesses its acceptability to family members. In this randomized controlled trial, 81 dyads (veteran/family member) were randomly assigned to the VIP or an enhanced usual care control condition. Randomization occurred after the baseline interview. Follow-up interviews occurred 3–4 months after baseline, and the interviewer was blinded to group assignment.	Family members in the VIP showed significantly lower depressive symptom scores and lower burden scores when compared to controls at follow-up. Satisfaction with caregiving did not differ between groups. Family members’ acceptance of the intervention was high. The VIP represents the first evidence-based intervention that considers both the veteran with TBI and the family, has a significant impact on family member well-being, and thus addresses a large gap in previous research and services for families of veterans with TBI.	Rehabilitation of physical and psychological impacts, as in Physical Impacts and Injuries; Stressors and Psychological Impacts
9. Angel CM, (2018). Team Red, White & Blue: a community-based model for harnessing positive social networks to enhance enrichment outcomes in military veterans reintegrating to civilian life. Transl Behav Med. 2018 Jul 17;8(4):554–564. [28]	Narrative	Military staff suffer diverse physical and psychological impacts compared to civilians but need to integrate into their new lives after leaving the service. They experience a reverse culture shock and difficulties adjusting and integrating into the new environment and civilian institutes and lack a cultivated and supportive social network, leading to greater morbidity and premature mortality, in part due to adopting risky health behaviors. This theory-based framework article describes how theoretical components can be translated into implemented social networks.	Based on the contributing evidence spanning many disciplines, from health services research to social psychology, and the vast experience of veterans, the authors defined an “enriched life” as being filled with health, people, and purpose. The model will benefit from longitudinal tests of effectiveness and formal testing of implementation-related factors. Future opportunities to partner with researchers and other organizations to understand program impact and identify effective intervention components are necessary.	Environmental exposure, education, and rehabilitation, as in Physical Impacts and Injuries; Environmental Exposure; Stressors and Psychological Impacts
10. Adirim, T. (2019). A military health system for the twenty-first century. Health Affairs, 38(8), 1268–1273. [29]	Narrative	This article portrays the drivers for the consolidation of the three medical departments—those of the Army, Navy, and Air Force—under one agency and reflects on the impacts of this transformation considering the Department of Defense’s (DoD) unique mission.	The structured rollout of the DoD’s consolidation plan allows for the creation and maturation of management and oversight functions to ensure that readiness is sustained and enhanced, culminating in an organizational model that allows for management and oversight of military medical health services delivery in support of combatant commander missions and the military’s requirements in an optimally efficient and effective manner. It also delivers safe, high-quality, and patient-centered care for all beneficiaries.	Collaborative health delivery, as in Physical Impacts and Injuries; Environmental Exposure; Stressors and Psychological Impacts
11. Tanielian, T. (2019). The US Military Health System: promoting readiness and providing health care. *Health Affairs*, *38*(8), 1259–1267. [10]	Narrative	This article supplies an overview of how the Military Health Service relies upon a specific program (TRICARE) to deliver both direct care and purchased care. The article also describes the history and evolution of the TRICARE program, presents information on the populations served and the volume and type of care rendered, and examines access and quality issues. Furthermore, it describes recent policy and operational changes that have influenced how the Military Health Service delivers health care, placing these changes in the context of other challenges facing the US healthcare system.	The ongoing geopolitical changes necessitate the full readiness of military healthcare for new physical, psychological, and environmental risks and their impacts. This will dramatically affect the nature and volume of care, as well as beneficiaries’ experiences with such care. Just as the Military Health Service continues to lead in innovation for the treatment of war-related injuries, it will also need to lead as a learning organization in adapting to a rapidly changing policy environment.	Collaborative and preventive health delivery, as in Physical Impacts and Injuries; Environmental Exposure; Stressors and Psychological Impacts
12. Cramm H, (2020). Impact of Canadian Armed Forces veterans’ mental health problems on the family during the military to civilian transition. Military Behavioral Health. 2020 Apr 2;8(2):148–58. [30]	Narrative	This study aimed to increase understanding of how veteran mental health problems impact the family during the transition to civilian life. A sequential, multi-qualitative design was used. Twenty-six family members of veterans with mental health problems completed individual interviews, and 9 took part in 3 focus groups. Veteran mental health problems created multifaceted and pervasive changes in family structure, roles, and routines, and these changes created negative mental health and well-being changes for family members. Transition may compound stressors related to mental health, with significant consequences for family systems.	Recent advances in technology have presented opportunities for mobile interventions to engage with caregivers who experience high levels of burden yet have little access to relevant mental health services. To the extent possible, the integration of families into assessment and intervention protocols for veterans living with mental health issues has been suggested. Rather than focusing on personal challenges, veterans may be more amenable to family-focused services, and relational factors are critical to the success of treatment following traumatic stress.	Collaborative and preventive health delivery, as in Physical Impacts and Injuries; Stressors and Psychological Impacts
13. Clausen, A. N. (2020). Combat exposure, posttraumatic stress disorder, and head injuries differentially relate to alterations in cortical thickness in military veterans. Neuropsychopharmacology, 45(3), 491–498. [31]	Quantitative comparative study	The present study examined the relationship between the severity of combat exposure and cortical thickness. Higher combat exposure is uniquely related to lower cortical thickness in the left prefrontal lobe and increased cortical thickness in the left middle and inferior temporal lobes, while PTSD is negatively related to cortical thickness in the right fusiform. Head injuries are related to increased cortical thickness in the bilateral medial prefrontal cortex. Combat exposure uniquely contributes to lower cortical thickness in regions implicated in executive functioning, attention, and memory after accounting for the effects of PTSD and prior head injuries.	These results highlight the importance of examining the effects of stress and trauma exposure on neural health in addition to the circumscribed effects of specific syndromic pathologies.	Physical and psychological impact of exposure, as in Physical Impacts and Injuries; Environmental Exposure; Stressors and Psychological Impacts
14. Gonzalez, J. A. (2021). The workplace integration of veterans: Applying diversity and fit perspectives. Human Resource Management Review, 31(2), 100775. [32]	Narrative review	This study briefly reviews relevant work from outside the management field and nascent work within the field to build a conceptual model for understanding the integration of veterans into the workplace, guiding future empirical research on veterans as human capital and their transition into civilian organizations as part of their societal reintegration, career development, and personal well-being.	Less is known about what happens when soldiers become civilians. This study may offer an applicable approach to studying military veterans and stimulate research and practice that contributes to organizational performance as well as veterans’ lives and well-being.	Social and workplace integration, as in Physical Impacts and Injuries; Environmental Exposure; Stressors and Psychological Impacts
15. Garvin LA, (2021). Interorganizational Care Coordination of Rural Veterans by Veterans Affairs and Community Care Programs: A Systematic Review. Med Care. 2021 Jun 1;59(Suppl 3):S259–S269. [33]	Systematic review	A review to examine the inter-organizational care coordination initiatives that Veterans Affairs (VA) and community partners have pursued in caring for rural veterans, including challenges and opportunities, organizational domains shaping care coordination, and among these, initiatives that improve or impede health care outcomes. Sixteen articles were included. Each captured a unique healthcare focus while examining common challenges. Four organizational domains emerged: policy and administration, culture, mechanisms, and relational practices. Exemplars highlight how initiatives improve or impede rural healthcare delivery.	Results provide exemplars of inter-organizational care coordination domains and program effectiveness. It suggests that partners’ efforts to align their coordination domains can improve healthcare, with rurality serving as a critical contextual factor. Findings are important for policies, practices, research, and community care partners who are committed to improving access and health care for rural veterans.	Health integration, as in Physical Impacts and Injuries; Environmental Exposure; Stressors and Psychological Impacts
16. Geretto, M. (2021). Occupational Exposures and Environmental Health Hazards of Military Personnel. *International Journal of Environmental Research and Public Health*, *18*(10), 5395. [1]	Review	This study describes the military staff’s exposure to environmental hazards. The exposures include sulfur mustard, organ chlorine, combustion products, fuel vapors, and ionizing and exciting radiation. Important factors to be considered are the lengths and intensities of exposures, their proximity to the sources of environmental pollutants, as well as confounding factors (cigarette smoke, diet, photo type, healthy warrior effect, etc.). Assessment of environmental and individual exposures to pollutants is crucial. Biomarkers of exposures and effects are tools to explore relationships between exposures and diseases in military personnel. Another major problem observed in this review is the lack of suitable control groups.	This review shows that only studies that analyzed epidemiological and molecular biomarkers in both exposed and control groups would provide evidence-based conclusions on exposure and disease risk in military personnel.	Environmental issues, as in Environmental Exposure
17. Gettings, R. (2022). Exploring the role of social connection in interventions with military veterans diagnosed with post-traumatic stress disorder: systematic narrative review. Frontiers in Psychology, 3646. [34]	Systematic review	This systematic review synthesized existing evidence incorporating elements of social connection, social isolation, and loneliness for military veterans with a diagnosis of PTSD and their impacts. Of the 28 collected studies, 11 directly addressed loneliness. Six themes were generated: rethinking the diagnosis of PTSD, holistic intervention, peer support, social reintegration, empowerment through purpose and community, and building trust.	A direct focus on social reintegration and engagement, psychological functioning, building trust, peer support, group cohesiveness, and empowerment through a sense of purpose and learning new skills may mitigate experiential loneliness and social isolation for veterans with PTSD. Further research is needed, specifically within communities.	Psychological support and treatment, as in Stressors and Psychological Impacts

## Data Availability

All data are included in the study. Specific requests can be answered and fulfilled by the lead author.

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
