# Peer review of "Supporting the Frontlines: A Scoping Review Addressing the Health Challenges of Military Personnel and Veterans"

_healthcare, 2023, doi:10.3390/healthcare11212870_

Round 1

Reviewer 1 Report

Comments and Suggestions for Authors

All my comments relate to the generality of the paper. I was excited by the title which promised to deliver much but the final result was confusing and disappointing. Clearly a scoping review attempting to address ALL the challenges of currently serving military and veterans would be enormous and encompass those still living from WW2. It would need to include all physical and mental health challenges encountered over 80 years and therefore to deliver just 17 papers suitable for review either meant the title is wrong or the wrong search terms have been used. Indeed it appears very odd that some of the numbered references did not appear in the scoping review as their topics and content would appear to sit in that space. I would be interested know why the authors thought over 400 papers were thought irrelevant, again suggesting the search terms were not appropriate. The paper falls between two stools, neither being a good scoping review (having too broad a theme) nor a good historical review, as the embedded scoping review merely confuses. I am left wondering from the conclusion if the authors are attempting to make a case for civilian health services to encompass care of military personnel if so much more needs to be made of comparing the US Veteran Administration structures with the UK NHS model and that currently in place in Ukraine where the two are not separated. If civilian military innovation is the key message then discussion on forward intensive care, renal replacement therapy, limb prosthetic development and patient transfer, among others, need expanding. In summary the paper needs reconstructing to deliver either a small and in my view a poorly constructed scoping review or a better more focussed review on the advancement of military health care in conjunction with civilian care. The latter would need to be time limited and probably focus on specific health care systems or geography as little discussed here would fit recent conflicts in Asia or Africa, nor internal insurgencies in S America.

Author Response

Dear Reviewer,

We wish to express our deepest gratitude for the thoughtful and detailed review of our manuscript. Your insights have been immensely valuable, providing us with clear guidance for refining our work. The depth and specificity of your feedback reflect a genuine commitment to ensuring the scientific robustness and clarity of articles, and we sincerely appreciate it.

Reviewer 2 Report

Comments and Suggestions for Authors

Summary

The article under review, "Healthcare Challenges for Military Personnel: A Scoping Review," explores the unique healthcare challenges faced by military personnel, both in terms of immediate injuries during combat and long-term medical needs, with a particular emphasis on veteran care. The authors employ a scoping review methodology to assess the current landscape of military healthcare, encompassing immediate medical responses and long-term healthcare strategies. The study utilizes systematic literature searches in various databases, employs a set of inclusion and exclusion criteria, and categorizes the results into thematic clusters related to physical impacts, environmental exposure, and stressors and psychological impacts.

General Comments

The article addresses an important and timely topic concerning the healthcare challenges faced by military personnel, especially veterans. The use of a scoping review methodology is suitable for the broad nature of the research question, and the article provides valuable insights into the various aspects of military healthcare. However, there are several areas where the article can be improved to enhance its quality and clarity.

Clarity and Structure

The article's structure could benefit from improved organization and clarity. The introduction section provides an adequate overview of the topic but is somewhat verbose. Consider condensing the introductory paragraphs and providing a more concise and focused introduction to the research question, objectives, and methodology.

Literature Search and Inclusion Criteria

The article mentions the use of four different databases and Google Scholar for the literature search, but it lacks information on the search terms and strategies used. It would be helpful to include the specific search terms and strategies employed to enhance the transparency and reproducibility of the study.

Inclusion and Exclusion Criteria

The article briefly mentions the inclusion and exclusion criteria but could provide more detail about the criteria used to select or exclude studies. This would help readers understand why certain studies were included while others were excluded.

Data Synthesis

The article mentions thematic categorization of the results into clusters related to physical impacts, environmental exposure, and stressors and psychological impacts. However, it would be beneficial to provide more detailed explanations and examples of the themes within each cluster to aid readers' comprehension.

Discussion

The discussion section is informative and provides valuable insights into the challenges and potential solutions for military healthcare. However, it lacks a clear summary of the key findings from the scoping review. It would be helpful to provide a concise summary of the main findings and their implications for military healthcare.

Language and Grammar

The article contains some language and grammatical issues that need attention. Proofreading and editing are required to improve the overall readability and professionalism of the manuscript.

Conclusion

The article addresses a significant topic related to healthcare challenges for military personnel, and its scoping review methodology is appropriate for the research question. However, there is room for improvement regarding clarity, structure, and transparency in the literature search process and data presentation. With revisions addressing these issues, the article has the potential to make a valuable contribution to the field of military healthcare.

Overall Recommendation

I recommend that the authors revise the manuscript to address the abovementioned issues and resubmit it for further review and consideration.

Author Response

Dear Reviewer,

Firstly, allow us to extend our sincerest appreciation for your thorough review of our manuscript. Each insight and observation you've shared not only showcases your profound expertise but has also offered us an opportunity to fortify and enhance our work.

Comment 1: Literature Search and Inclusion Criteria
Observation: The use of multiple databases, including Google Scholar, is noted. However, specific search terms and strategies employed could be made more transparent for replicability purposes.
Response: We fully recognize the importance of clarity and precision in detailing our search strategy. In response to this observation, we've meticulously detailed our search terms and strategies to ensure transparency. To see these comprehensive updates, we direct you to points 2.1 through 2.8 and Table 1 in the method section of our revised manuscript.

Comment 2: Inclusion and Exclusion Criteria
Observation: The criteria for selecting studies could be elucidated further to help readers grasp the rationale behind including some studies and excluding others.
Response: Your point highlights an essential facet of research transparency. We have now expanded upon the inclusion and exclusion criteria, providing more granular details about the parameters that guided our selection. For a deeper understanding, please refer to points 2.1 through 2.8, including Table 1, in the method section of the amended manuscript.

Comment 3: Data Synthesis
Observation: While the thematic categorization of results into specific clusters is noted, a deeper dive into these themes would enhance readers' comprehension.
Response: We concur with your observation. To address this, we've enriched our explanation concerning the thematic clusters, detailing the rationale and examples behind each category. In the revised manuscript, we invite you to consult point 2.8 in the method section and table 2. Here, we've elucidated how each study was themed and then aggregated into distinct clusters. The in-depth categorizations include: 1) Physical Impacts and Injuries; 2) Environmental Exposure and Impact; and 3) Stressors and Psychological Impacts. These can be readily referenced in the last column of table 2.

To conclude, your thorough critique has been a cornerstone in our revision process, ensuring our research is both rigorous and accessible. We remain deeply grateful for your continued guidance and expertise in this endeavor.

Warmest regards,

in this process.

Authors

Reviewer 3 Report

Comments and Suggestions for Authors

Thank you for the opportunity to review this paper. The study focuses on an interesting topic. However, I have the following comments.

·        Justification is poor. It is not clear why the study is needed. It is also unexpected that no other review has been conducted on the matter.

·        Scoping review has been used and in the paper it is presented as it is a combination of systematic search with non-systematic review, which does not make much sense or raises questions. Is there such information in the PRISMA tailored for scoping reviews? The authors should better explain why a scoping review is more suitable for their study.

·        Google scholar is presented as a database like Scopus is. This is wrong as google scholar is only a search engine and it is used only as supplementary source.

·        The study is about health challenges. However, in the search strings keywords, such as healthcare needs, disease, illness, management have not been included. It is puzzling why they were left out. The authors should explain why they have included only the keywords they outline in their paper and not others. The key thing with all research, including reviews, is that everything should be justified.

·        What is “non-systematic review” of the selected articles? What did you exactly and why what you did was not systematic?

·        Have you used a double-review process for selecting and reviewing the papers?

·        Have you critically appraised the articles based on published criteria? If not, why not? If yes, then please include the appraisal table citing the criteria.

Comments on the Quality of English Language

Another check/ edit is needed to ensure better flow. 

Author Response

Dear Reviewer,

We sincerely appreciate the positive feedback regarding the relevance, timeliness, structure, and clarity of our paper. It is encouraging to know that the core content and the presentation resonate well with the reader.

         Comment 1: Scoping review has been used and in the paper it is presented as it is a combination of systematic search with non-systematic review, which does not make much sense or raises questions. Is there such information in the PRISMA tailored for scoping reviews? The authors should better explain why a scoping review is more suitable for their study.

Response 1: Thank you for your insightful comment. The reason for choosing a Scoping Review and why we called our review non-systematic has been clarified in the method section, and relevant references have been given. Please see, point 2.1 and forward.

The reason why we called our review, non-systematic, are clarified under the point 2.7, page 5.

Comment 2: Google scholar is presented as a database like Scopus is. This is wrong as google scholar is only a search engine and it is used only as supplementary source.

Response 2: As you correctly mentioned, Google Scholar is not a database and should be sorted as other sources. We have made changes in Figure 1 to clarify and emphasize your comment.

Comment 3: The study is about health challenges. However, in the search strings keywords, such as healthcare needs, disease, illness, management have not been included. It is puzzling why they were left out. The authors should explain why they have included only the keywords they outline in their paper and not others. The key thing with all research, including reviews, is that everything should be justified.

         Response 3: Thank you again for your insightful comment. In point 2.2, in the revised version of the paper, we have explained why and how the keywords were used. We also tested additional keywords with insufficient outcomes.  

Comment 4: Comment 4: What is “non-systematic review” of the selected articles? What did you exactly and why what you did was not systematic? A systematic review

Response 4: The majority of the studies were narrative reviews (n=11), a few were systematic reviews, one was a randomized trial, one was registry-based, and another comparative study. Consequently, the evidence-based quality of the included 17 studies differed and did not allow a proper assessment. This has been added to the revised version, page 6, last paragraph. The choice of method is also discussed in detail in the method section, point 2.1.

Comment 5: Have you used a double-review process for selecting and reviewing the papers?

Response 5: The review process is outlined in section, 2,7, page 5.

Comment 6: Have you critically appraised the articles based on published criteria? If not, why not? If yes, then please include the appraisal table citing the criteria.

Response 6: All abstracts and if necessary full texts have been studied and evaluated by two reviewers. Please see the revised method section.

Once again, we are grateful for the comprehensive feedback. These insights were instrumental in refining our paper, and we believe that the adjustments made have enhanced its quality and readability. We hope the revised manuscript will meet the standards of the journal.

Warm regards,

Authors

Round 2

Reviewer 1 Report

Comments and Suggestions for Authors

I remain unconvinced by this paper which contains internal inconsistensies and the comments or explanation in the accompanying letter do not fully answer initial critique. The authors state they do not search for papers on specific or single diagnosis but the bibliography they settle on (just 17 papers) include papers 6 and 8 focusing entirely on traumatic brain injury, papers 2 and 4 specific to PTSD and even more niche paper 13 looking at cortical changes. These seem very singular despite the authors protestations they were NOT investigating a single condition, in their own words 'without any emphasis on a single medical condition' - lines 89 & 90. Again I see little connection between a poorly conducted/constructed scoping review and the general review nature of the paper itself. Even the abstract describes the scoping review to look at 'current landscape' (line 25) with the next statement (line 26) stating there has been a change in injury profile 'over time'. The scoping review and discussion of it can not be both current and a review over time. The authors may suggest by over time they describe the time scale of the papers, 2009 - 2022 but much of the discussion in the paper under review refers to WW2 and Vietnam . Their own limitations section provides sufficient additional evidence to reject this paper.

Author Response

Response to Reviewer Comments

Dear Reviewer,

Thank you for your insightful comments and thorough review of our manuscript. We genuinely appreciate the time and effort you invested in evaluating our paper. In response to your concerns, we would like to address each point you raised:

  1. Scope of Selected Papers: We recognize the concerns regarding papers 6 and 8, which focus on traumatic brain injury, papers 2 and 4 specific to PTSD, and paper 13 exploring cortical changes. Our intention in this scoping review was not to exclude papers focusing on a specific condition, but rather not to have the emphasis of our entire review hinge on one single condition. The inclusion of these papers is reflective of the fact that some conditions or findings might be particularly pertinent to understanding care dynamics of the special population under review, even if they are condition-specific.

  1. Connection Between Scoping Review and General Review: The essence of our paper is to bridge the gaps between a general review and a more pointed scoping review. While a scoping review serves to explore the breadth of the field, our manuscript attempts to provide depth in certain areas, complementing the broader view with specific instances or conditions that have major implications for care.

  1. Temporal Discrepancies in the Review: We understand the perceived inconsistency between examining the "current landscape" and discussing changes "over time". Our aim was to provide a comprehensive overview of the landscape from 2009-2022, while also drawing on historical contexts like WW2 and Vietnam to offer perspective on how certain aspects of care or understanding have evolved. These historical references provide context rather than primary evidence for our main arguments.

Regarding the limitations section, we believe it's crucial to transparently acknowledge potential shortcomings in any academic endeavor. The points raised therein are intended to guide readers in their interpretation of our findings, rather than discrediting the methodology or results entirely.

In light of the above explanations, we kindly request reconsideration of our manuscript. We would also like to point out that two other reviewers have endorsed our paper and appreciated the changes we made based on their feedback. Additionally, the Academic Editor has expressed a favorable view of our work. While we understand and respect the diversity of opinions in the peer review process, we believe that the collective feedback indicates the merit of our paper.

We value your insights and hope that our clarifications address your concerns. Our team is committed to enhancing the quality of our paper and ensuring that it provides meaningful insights to the academic community. We hope that you would consider our perspectives and reassess the potential contribution of our work.

Thank you for your attention and consideration.

Warm regards,

Authors

Reviewer 2 Report

Comments and Suggestions for Authors

This research reached the quality of our research.

Author Response

Dear Reviewer,

Thank you for your time and efforts to enhance our research quality.

Reviewer 3 Report

Comments and Suggestions for Authors

It seems that my comments have been addressed. 

Comments on the Quality of English Language

No comments.

Author Response

(The authors gave the same response as above.)
